# Recent and Future Strategies to Overcome Resistance to Targeted Therapies and Immunotherapies in Metastatic Colorectal Cancer

**DOI:** 10.3390/jcm11247523

**Published:** 2022-12-19

**Authors:** Giulia Dazio, Samantha Epistolio, Milo Frattini, Piercarlo Saletti

**Affiliations:** 1Laboratory of Molecular Pathology, Institute of Pathology, Ente Ospedaliero Cantonale (EOC), 6601 Locarno, Switzerland; 2Department of Medical Oncology, Clinica Luganese Moncucco, 6900 Lugano, Switzerland

**Keywords:** gene mutations, gene fusions, targeted therapies, resistance, immunotherapies, microsatellite instability

## Abstract

Colorectal cancer (CRC) is the third most common cause of cancer-related deaths worldwide, and 20% of patients with CRC present at diagnosis with metastases. The treatment of metastatic CRC is based on a fluoropyrimidine-based chemotherapy plus additional agents such as oxaliplatin and irinotecan. To date, on the basis of the molecular background, targeted therapies (e.g., monoclonal antibodies against epidermal growth factor receptor or inhibiting angiogenesis) are administered to improve the treatment of metastatic CRC. In addition, more recently, immunological agents emerged as effective in patients with a defective mismatch repair system. The administration of targeted therapies and immunotherapy lead to a significant increase in the survival of patients; however these drugs do not always prove effective. In most cases the lack of effectiveness is due to the development of primary resistance, either a resistance-inducing factor is already present before treatment or resistance is acquired when it occurs after treatment initiation. In this review we describe the most relevant targeted therapies and immunotherapies and expand on the reasons for resistance to the different approved or under development targeted drugs. Then we showed the possible mechanisms and drugs that may lead to overcoming the primary or acquired resistance in metastatic CRC.

## 1. Introduction

Colorectal cancer (CRC) is the third most common cause of cancer-related deaths worldwide, with more than 1.85 million cases and 850,000 deaths annually [1]. In the United States, 151,030 new cases and 52,580 deaths are estimated for 2022 [2]. Among people diagnosed with CRC, 20% present with metastatic CRC (mCRC), and 40% develop recurrence after curative surgery [3,4]. The 5-year survival rate for patients affected by mCRC is poor and accounts for less than 20% [1]. The systemic treatment of patients with mCRC is based on a fluoropyrimidine [5-fluorouracil (5-FU) or the prodrug capecitabine]-based chemotherapy plus additional agents, either as doublets or triplets, encompassing oxaliplatin (OX) and irinotecan (IRI). Moreover, monoclonal antibodies (MoAbs) inhibiting either angiogenesis [anti-vascular endothelial growth factor (VEGF), such as bevacizumab, aflibercept or ramucirumab] or epidermal growth factor receptor (EGFR) (cetuximab or panitumumab) can be considered in addition to chemotherapy. More recently, immunological agents emerged as effective in patients with defective mismatch repair systems. Figure 1 summarizes all the available options.

In selected patients, resection of metastases can be considered based on the tumor biology and patient’s characteristics [5].

Here, we review systemic therapy, focusing on mechanisms of resistance of targeted and immunotherapy agents. Finally, efforts to overcome resistance are described.

## 2. Drug Resistance in Cancer

Drug resistance (DR) is a crucial aspect in cancer care. Although multi-DR is a well described phenomenon in medicine, this topic has become increasingly relevant with the implementation of molecularly targeted agents in patients with cancer. Different, interplaying factors (i.e., the pharmacological properties of the compound, molecular characteristics of cancer cells and the extrinsic factors cumulatively) account for DR, influencing the clinical outcome [6,7]. DR mechanisms can be classified as intrinsic (or primary) if the resistance-inducing factor is already present before treatment or acquired if DR occurs after treatment initiation. However, the boundary between the two definitions is faint [8]. Indeed, the administration of targeted therapies may confer an initial benefit (by killing the clones harbouring the specific target), but then other minor clones with different molecular characteristics, present at the beginning but undectectable by standard techniques, may take over, having DR as a consequence [8].

Biological determinants of DR are tumor heterogeneity, physical barriers (e.g., the blood-brain barrier), immune system, tumor environment, tumor growth, tumor burden and undruggable genome [6].

## 3. Resistance Mechanisms to Targeted Agents in mCRC

Targeted therapies have represented a major achievement in the modern treatment of cancer, translating in a significant increase in overall survival (OS) and quality of life. However, several issues have emerged, and the mechanisms of resistance are among the most challenging.

A summary of the resistance mechanisms related to targeted therapies and immunotherapy and of the possibilities to overcome resistance is reported in Table 1.

### 3.1. MoAbs against ErbB/HER Family

The epidermal growth factor receptor (EGFR) and the human epidermal growth factor 2 (HER2) belong to the ErbB/HER family, which consists of four transmembrane glycoprotein tyrosine kinase receptors with an analogous structure consisting of an extracellular ligand-binding domain, a single hydrophobic transmembrane region and an intracellular segment that contains a conserved tyrosine kinase domain. These receptors are called ErbB1 (EGFR/HER1), ErbB2 (Neu/HER2), ErbB3 (HER3) and ErbB4 (HER4) [9,10]. Their functionality is based on a homo- or heterodimerization with other members of the same family, also mediated by the specific ligand (with the exclusion of HER2, which is a ligand orphan), activating intracellular signaling pathways including RAS proteins/RAF/mitogen-activated protein kinase (MEK)/ERK and PI3K/AKT/mTOR. Once activated, these pathways promote cell growth, survival and cellular migration [9].

In CRC, EGFR and HER2 are overexpressed in 59–85% and 2–11% of patients, respectively [5,11]. As EGFR and HER2 are the family members with the best-defined roles in mCRC, most clinical trials have incorporated compounds targeting these receptors.

#### 3.1.1. Anti-EGFR MoAbs

Cetuximab (Erbitux^®^; Merck, Readington Township, NJ, USA) was the first targeted drug approved for the treatment of mCRC (Figure 1). Cetuximab is a recombinant chimeric human/mouse IgG1 MoAb that competitively binds EGFR, inhibiting the binding of epidermal growth factor [12]. In addition, it induces EGFR internalization and degradation once bound to the external domain of EGFR [13].

The first preclinical data with anti-EGFR compounds were announced in 1995 [13]. The Food and Drug Administration (FDA) and the European Medicines Agency (EMA) approved cetuximab in combination with IRI in 2004 [5], based on both the pivotal European randomized BOND study and two clinical trials conducted in the United States [14]. The approval was given as a consequence of improved time to progression (TTP) with cetuximab plus IRI in patients with mCRC overexpressing EGFR, after failure with IRI [14].

Nowadays, the EMA and FDA recommend cetuximab for the treatment of patients with mCRC without mutations in RAS genes [mainly Kirsten rat sarcoma virus (KRAS) and neuroblastoma RAS viral (v-ras) oncogene homolog (NRAS)]. In left-sided tumors cetuximab can be administered as a first-line treatment in combination with IRI- or OX-based chemotherapy [folinic acid, 5-FU and IRI (FOLFIRI) or folinic acid, 5-FU and OX (FOLFOX)] or as monotherapy in patients refractory to OX- and IRI-based regimens [15,16].

The randomized phase III CRYSTAL trial showed that cetuximab plus FOLFIRI correlated with improved PFS [8.9 mo vs. 8.0 mo, hazard ratio (HR): 0.85, *p* = 0.048] compared to FOLFIRI, although the OS was not significantly affected (HR: 0.93, *p* = 0.31) [17]. An updated survival analysis of the CRYSTAL trial, including an increased number of patients analyzed for tumor mutation status, demonstrated that the addition of cetuximab to FOLFIRI as first-line therapy improves survival in patients with KRAS wild-type (wt) mCRC [18]. In addition, a phase III trial reported a higher response rate (RR) favoring cetuximab and IRI compared to single agent cetuximab (22.9% vs. 10.8%, *p* = 0.007) [12,19].

The murine-human chimeric antibody cetuximab can also induce immunogenic infusion reactions. As a consequence, the fully humanized antibody panitumumab (Vectibix^®^; Amgen, Thousand Oaks, CA, USA) (Figure 1) has been developed to overcome these adverse events. The efficacy of this MoAb has been evaluated in the PRIME trial, in which FOLFOX plus panitumumab resulted in a better PFS (10.0 mo vs. 8.6 mo, HR 0.80, *p* = 0.01) compared to FOLFOX alone in KRAS wt cases [20,21]. In addition, as expected, panitumumab correlated with a lower risk of hypersensitivity reactions (0.6–3.0%) compared to cetuximab (3.5–7.5%) [22].

The efficacy of the maintenance strategy with panitumumab plus 5-FU/folinic acid was evaluated in a retrospective analysis of the PRIME and PEAK trials, showing a better PFS and OS than 5-FU alone [23]. In addition, in the PANAMA trial the addition of panitumumab to 5-FU resulted in a significantly superior PFS compared to 5-FU/folinic acid alone (8.8 mo vs. 5.7 mo, HR: 0.72, 95% confidence interval: 0.60 to 0.85, *p* = 0.014) [24]. Currently, the FDA and EMA recommend panitumumab for the first-line treatment of RAS genes wt mCRC with FOLFOX or as monotherapy after failure of fluoropyrimidine, OX and IRI-containing regimens [25,26].

For second-line treatment, both cetuximab and panitumumab failed to show improved PFS or OS. The EMA suggests panitumumab administration in combination with FOLFIRI in patients with wt RAS genes mCRC who have received first-line fluoropyrimidine-based chemotherapy (excluding IRI) [26].

#### 3.1.2. Anti-EGFR MoAbs Resistance

Only 10% of chemo-refractory patients achieve objective response to both cetuximab and panitumumab as single agents [27,28]. The first explanation for primary resistance to anti-EGFR drugs can be the presence of mutations in RAS genes [29,30], which are found to be mutated in nearly half of patients with CRC (36% for KRAS and 10% for NRAS, respectively) [31] (Figure 2). Indeed, mutant KRAS proteins lead to hyperactivate the main EGFR downstream pathway, resulting in ineffectiveness of anti-EGFR MoAbs. Despite this, that literature reports that not all the patients harboring KRAS mutations developed resistance to anti-EGFR therapies because resistance correlates with the specific mutation detected in KRAS [32,33].

However, only a subgroup of patients with both EGFR overexpression and KRAS exon 2 wt tumors experienced a real benefit from anti-EGFR therapies. As a consequence, additional studies were conducted on EGFR/HER2 and related downstream pathways.

As far as HER2 amplification is concerned, this alteration has been proposed as a possible factor for resistance although currently it remains unclear whether amplification can affect efficacy of anti-EGFR therapy (Figure 2) [34,35]. In the PRESSING (primary resistance in RAS genes and BRAF wt patients with mCRC treated with anti-EGFR MoAbs) panel, Cremolini et al. [36] identified several uncommon genomic alterations including HER2 amplification/activating mutations, MET amplification (Figure 2), ROS1/neurotrophic tyrosine receptor kinases (NTRK1-3)/RET rearrangements, PIK3CA exon 20, PTEN and ALK mutations. Among these, HER2 overexpression/amplification or, less frequently, HER2 mutations are the most common ones. Some studies suggest that HER2 amplification may be a predictor of a shorter PFS in patients treated with cetuximab [37,38]. In particular, Martin et al. [38] identified three distinct subgroups of patients by correlating HER2 gene copy number (evaluated by fluorescent in situ hybridization) with the anti-EGFR MoAbs response and outcome. In the first group, patients with HER2-amplified tumors and treated with anti-EGFR MoAbs exhibited shorter OS and PFS. On the contrary, in the remaining two groups including patients with increased HER2 gene copy number (due to polysomy) or HER2 gene amplification in minor clones were more likely to respond to anti-EGFR agents and exhibit a better prognosis. Another study demonstrated that survival of amplified HER2 in patients treated with anti-EGFR MoAbs is similar to patients with a RAS gene mutation [39].

Retrospective cohorts showed that MAPK pathway alterations can account for primary resistance to single agents cetuximab or panitumumab in chemo-refractory mCRC [36,37,40,41] (Figure 2). Indeed, mutations in KRAS exons 3–4 and NRAS exons 2, 3 and 4 emerged as crucial for anti-EGFR MoAb resistance even when administered in combination with chemotherapy [42,43,44]. On the contrary, BRAF V600E mutation seems to predict resistance only when anti-EGFR MoAbs are used as single agent not in combination with chemotherapy [42]. However, based on the analyses of the ARCAD study, anti-EGFR treatment should not be recommended in BRAF mutant cases in any situation [45].

PIK3CA exon 20 mutations and PTEN loss have also been analyzed as potential factors in primary resistance to anti-EGFR MoAbs due to their role in constitutive activation of other EGFR downstream pathways (Figure 2) [32,41,46,47]. However, to date data are controversial, especially regarding PTEN loss [5,46,48,49] and further studies are warranted.

Concerning EGFR activation, it has been proposed that expression of amphiregulin (AREG) and epiregulin (EREG), two EGFR-specific ligands, may be relevant in predicting anti-EGFR response. In a prospective clinical trial, Khambata-Ford et al. [50] found that AREG and EREG gene expression levels measured before treatment were associated with cetuximab efficacy (Figure 2). Indeed tumors expressing high levels of AREG and EREG were more likely to respond to cetuximab (EREG *p* = 0.000015 and AREG *p* = 0.000025) in comparison to patients showing low expression of the ligands [50]. A later study confirmed the same results in KRAS wt cases [51]. These two publications demonstrate that high AREG and EREG gene expression levels may represent additional predictive biomarkers of response to anti-EGFR therapies, but so far they have not been introduced into clinical practice [50,51].

Another mechanism that could influence anti-EGFR efficacy is epithelial-to-mesenchymal transition. Previous studies suggested that this phenomenon occurring in cancer cells may attenuate the role of EGFR signaling by regulating cell proliferation and survival [52]. The correlation between epithelial-to-mesenchymal transition and resistance to erlotinib (an EGFR tyrosine kinase inhibitor effective in the presence of EGFR activating mutations) has been demonstrated in non-small cell lung cancer. However, this finding was not observed in mCRC [53].

In addition to primary anti-EGFR resistance, patients with mCRC exposed to these compounds can develop, during or after the treatment, new cell clones accounting for an acquired resistance. The most common mechanism driving secondary resistance to anti-EGFR includes the same genetic alterations previously described. For example, secondary KRAS and NRAS mutations in exons 3 and 4 were found to emerge in tumor biopsies and in circulating tumor DNA from patients who progressed to anti-EGFR MoAbs [54,55,56,57]. In addition, BRAF and PIK3CA mutations were also found in patients who experienced disease progression after anti-EGFR exposure [58].

#### 3.1.3. Overcoming Resistance to Anti-EGFR MoAbs

The most intuitive approach in order to block anti-EGFR MoAbs resistance and possibly restore efficacy is the inhibition of the pathways involved in the altered markers leading to resistance. These altered genes can be targeted either simultaneously or sequentially. For example, in cell lines exhibiting BRAF V600E mutation, KRAS wt and EGFR gene amplification, Di Nicolantonio et al. [59] showed that cetuximab alone was substantially ineffective, but sensitivity was restored by combining cetuximab and sorafenib, an inhibitor of the BRAF V600E. In addition, the IMPROVEMENT phase II trial demonstrated, in a small cohort, promising antitumour activity in BRAF V600E mutated patients treated with vemurafenib (see Section 3.2.1) and cetuximab in combination with FOLFIRI [60].

Another way to overcome resistance to anti-EGFR therapies is by targeting crucial EGFR downstream effectors such as mitogen-activated protein kinase (MEK or MAP2K) or mechanistic target of rapamycin (mTOR). The inhibition of these genes (dual EGFR/MEK and/or mTOR) resulted in a better response in tumor models exhibiting RAS family, BRAF or PIK3CA mutations [61,62,63,64].

Two clinical trials evaluated the strategy of rechallenge with anti-EGFR MoAbs in third-line treatment after a previous response [41]. Alongside this finding, other studies showed that cetuximab sensitivity might be restored in RAS genes/BRAF wt patients with mCRC who either acquired resistance to cetuximab-based therapy in first-line therapy [65] or exhibiting RAS genes mutations in those who received bevacizumab-containing regimens for first-line treatment [66]. These latest studies raise the hypothesis that anti-angiogenic agents (e.g., bevacizumab, described in particular in the following section named “Targeting angiogenesis”) could revert tumors from RAS genes mutant to RAS genes wt status by selectively (or at least predominantly) killing RAS mutant clones, which are more angiogenesis-dependent than RAS genes wt cells. This finding may theoretically open up the possibility to treat de novo or expose anti-EGFR MoAbs-treated patients with previously demonstrated RAS genes mutant status. This hypothesis seems to be confirmed in both tissue and plasma samples [65,66,67], but it deserves further investigation.

Second-generation anti-EGFR MoAbs have been developed. The engineered molecules both induce antibody-mediated cytotoxicity and increase receptor internalization. Imgatuzumab (GA201^®^; Roche, Basel, Switzerland) exhibited in vivo efficacy in a phase I study. However, in a multicenter randomized phase II study, imgatuzumab in combination with FOLFIRI failed to produce a survival advantage for second-line treatment compared to FOLFIRI +/− cetuximab. This finding was similar in KRAS mutant or wt cases [68]. Another second-generation synergistic MoAb, Sym004 (futuximab + modotuximab; Symphogen, Copenhagen, Denmark), showed promising activity in a preclinical study [69], which unfortunately in a phase I trial did not translate at a clinical level, as only 17 out of 39 patients with mCRC with acquired resistance experienced tumor shrinkage [41].

#### 3.1.4. Anti-HER2 MoAbs

HER2 shares the same pathway as EGFR, and HER2 overexpression or mutations have been proposed as a putative mechanism of resistance to the anti-EGFR MoAbs. HER2 targeted therapies, including the fully humanized MoAb trastuzumab (Herceptin^®^; Roche) (Figure 1), pertuzumab (Perjeta^®^; Genentech, Roche, South San Francisco, CA, USA) (Figure 1) or the small-molecule tyrosine kinase inhibitors (TKIs), such as the dual EGFR/HER2 TKI lapatinib (Tyverb^®^; GlaxoSmithKline, Brentford, UK) and neratinib (Nerlynx^®^; Pierre Fabre, Paris, France), are approved in the treatment of breast and gastric cancers. However, there is no robust evidence of efficacy of HER2 blockade in mCRC [70].

A proof-of-concept study in HER2-amplified cetuximab-resistant CRC xenograft models found that pertuzumab given alone or in combination with cetuximab induced a negligible delay in tumor growth. Conversely, the combination of lapatinib and pertuzumab or lapatinib and cetuximab induced a significant and durable tumor shrinkage [35]. The synergistic antiproliferative effect was demonstrated in cetuximab-resistant CRC cell lines, treated either with trastuzumab plus lapatinib or trastuzumab plus cetuximab [71,72]. These preclinical data constituted the basis for clinical trials in patients with mCRC. First, the project HERACLES, including a group of small studies and evaluating the administration of anti-HER2 MoAbs (i.e., trastuzumab and pertuzumab) plus cetuximab or cytotoxic agents (i.e., IRI, 5-FU and OX), which was prematurely closed because of the severe overlapping toxicities [73] or poor accrual [74]. Subsequently, phase I basket trials including 6 HER2-positive patients with refractory CRC who reported no response to administration of paclitaxel, interleukin (IL)-12 and trastuzumab [75].

More recently one study (HERACLES-A) investigated chemotherapy-free regimens based on the combination of HER2-targeted drugs [76]. In the HERACLES-A study, including patients with mCRC with wt KRAS exon 2 (codons 12 and 13)/HER2-positive tumors and refractory to standard treatments including cetuximab or panitumumab, Sartore-Bianchi et al. [76] reported an objective RR of 30% after exposure to trastuzumab plus lapatinib, with a median duration of response of 9.5 mo and a median PFS of 5.2 mo. The encouraging efficacy data and the manageable toxicity profile (no grade-4 or grade-5 adverse events) represented an option in heavily pretreated patients with mCRC [76].

The intriguing results of this trial brought to the development of new HER2 inhibitors or new methodologies to block HER2, among these TKIs (e.g., tucatinib (Tukysa^®^; Seagen, Bothell, WA, USA)), HER2-targeted antibody-drug conjugates (e.g., A166, DS-8201, TDM-1, ZW25 and ZW49) and HER2-targeted immunotherapy (i.e., vaccines, donor-derived natural killer cells and CAR-T cells) [70,77]. The most relevant data involved trastuzumab deruxtecan and tucatinib. In the DESTINY-CRC01 phase II study, Siena et al. [78] found a promising activity of trastuzumab deruxtecan in mCRC HER2-positive patients refractory to standard treatments including anti-EGFR MoAbs. Concerning tucatinib, a single arm phase II trial (MOUNTAINEER trial) is ongoing with the purpose of evaluating this selective tyrosine inhibitor of HER2 in combination with trastuzumab in patients with HER2-positive, RAS genes wt mCRC who have previously received 5-FU, OX, IRI and an anti-VEGF MoAbs [79]. The results have been presented at the European Society for Medical Oncology World Congress on Gastrointestinal Cancer (July 2022). With a median follow up of 20.7 mo, an objective RR of 38.1% was reported in those who received the combination, with median duration of response of 12.4 mo, median PFS of 8.2 mo and OS of 24.1 mo.

#### 3.1.5. How to OVERCOME Anti-HER2 Resistance?

As the clinical experience of anti-HER2 blockade in mCRC is limited, robust evidence on resistance to these therapies is lacking. The main mechanism of resistance to anti-HER2 agents seems to be the occurrence of mutations in downstream pathways, as for EGFR (Figure 3).

Siravegna et al. [80] investigated anti-HER2 resistance mechanisms by analyzing the evolution of single metastases during treatment in order to discover temporally possible determinants of resistance. Indeed, the authors found RAS family and PIK3CA mutations in the circulating tumor DNA of the majority of refractory patients to HER2 inhibitors [80].

Although the mechanisms underlying resistance to HER2 inhibitors are poorly understood, from a clinical point of view it has been reported that trastuzumab plus lapatinib can overcome resistance in patients previously treated with pertuzumab plus trastuzumab [22].

### 3.2. BRAF Inhibitors

Mutations in the BRAF gene, besides being a mechanism of resistance to anti-EGFR and anti-HER2 agents, represents a potential therapeutic target. The serine/threonine protein kinase BRAF belongs to the RAF family [81,82], and it is activated by RAS family members in response to signals upon activation of receptor tyrosine kinases [82,83]. BRAF plays a role in MAPK pathway activation, and it is involved in cellular growth, proliferation and differentiation [84]. BRAF mutations are found in approximately 10% of patients with mCRC and in general are associated with microsatellite instability (MSI), hypermethylation, and minimal chromosomal instability [83,85,86,87].

The most diffused BRAF mutation, represented by a transversion in the nucleotide 1799 leading to the valine to glutamate change (V600E) is usually observed in right-sided tumors [88] and is associated with a significantly poorer prognosis compared to BRAF wt tumors, with a median survival of approximately 12 mo [89,90,91,92]. BRAF V600E mutation is present in 8% to 10% of patients with mCRC and is significantly more common in MSI-high (MSI-H) individuals (38.9%) than in MSI-low (MSI-L) individuals (9.3%) [32,59,83,84,90,91,92]. Furthermore, it has been demonstrated that BRAF V600E mutated CRCs metastasize more frequently in the peritoneum and receive less benefit from standard treatment [89,93].

#### 3.2.1. Anti-BRAF Agents

Vemurafenib (Zelboraf^®^; F. Hoffmann-La Roche/Genentech, San Francisco, CA, USA) (Figure 1) is a selective TKI specific to the ATP-binding domain of BRAF V600E [94]. Since BRAF V600E mutation results in the constitutive activation of the MAPK pathway, the assumption is that vemurafenib could decrease BRAF and downstream pathway activity. Vemurafenib represents a valuable therapy in multiple advanced malignancies characterized by BRAF V600E mutation. For instance, in metastatic melanoma vemurafenib was associated with high RR exceeding 50% and prolonged PFS [95,96,97]. In mCRC harboring the BRAF V600E mutation treated with vemurafenib monotherapy, the RR was only 5% [98]. Another BRAF inhibitor recently received FDA approval [99,100]. Encorafenib (Braftovi^®^; Array BioPharma Inc., Boulder, CO, USA) (Figure 1) is an RAF TKI administered in combination with cetuximab for the treatment of BRAF V600E patients with mCRC who have received prior therapy [101]. Compared to other approved BRAF inhibitors, encorafenib has prolonged pharmacodynamic activity [102], and pivotal clinical data were recently updated [103].

#### 3.2.2. Resistance to BRAF Inhibitors

Preclinical studies showed that different sensitivity of BRAF V600E inhibition in melanoma and CRC may be due to the robust adaptive feedback signaling networks harbored in CRC, which leads to the reactivation of MAPK signaling following treatment with a BRAF inhibitor. The inhibition of BRAF V600E causes a reduction in MAPK signaling, which leads to the loss of the ERK-dependent negative feedback. As a consequence, RAS family activity will increase by bypassing the effects of the BRAF inhibitor upon generating BRAF inhibitor-resistant RAF dimers. The increase in RAS genes activity is thought to be driven primarily by receptor tyrosine kinase signaling, in particular EGFR, which is more activated in CRC than in melanoma. Overall, this process will restore the MAPK pathway signaling even upon BRAF pharmaceutical inhibition (Figure 4) [104,105].

The limited efficacy of BRAF inhibitors given as single agents in patients with BRAF-mutant CRC led to investigations of the combination of BRAF and EGFR inhibitors, demonstrating improved activity [106,107,108]. Moreover, preclinical studies suggested that combined inhibition of BRAF and MEK were more effective compared to BRAF and anti-EGFR inhibition [104,109,110]. In addition, another possible mechanism of resistance to BRAF-targeted monotherapy is the increased activation of the PI3K/AKT [111,112].

#### 3.2.3. Overcoming Anti-BRAF Resistance

Studies conducted on mice showed a tumor regression after dual blockade of BRAF and EGFR using vemurafenib and cetuximab. Clinical activity of BRAF and EGFR inhibition was also demonstrated in pilot studies [113,114]. In addition, preclinical studies in patient-derived xenografts demonstrated that IRI in combination with dual BRAF and EGFR inhibition enhanced antitumor efficacy [115]. These findings led to a phase Ib study in which vemurafenib plus IRI and cetuximab were given in BRAF V600E mutated patients with mCRC. A RR of 35% was reported, together with a promising PFS [106]. Based on these findings, a prospective randomized phase II trial of cetuximab and IRI with or without vemurafenib was conducted, reporting a significant improvement in PFS (HR: 0.50, *p* = 0.001) in patients treated with a vemurafenib-containing regimen [116]. Early clinical trials showed promising results with encorafenib in combination with cetuximab [107,117]. The BEACON CRC study compared encorafenib plus cetuximab with or without the MEK inhibitor binimetinib (Mektovi^®^; Array Biopharma) vs. IRI or FOLFIRI plus cetuximab in previously treated patients with the BRAF V600E mutation. Regardless of binimetinib, encorafenib plus cetuximab was superior in terms of OS (HR: 0.61) and overall RR compared to the reference arm [103,118,119].

Cetuximab and encorafenib plus alpelisib (Piqray^®^; Novartis Pharmaceuticals, East Hanover, NJ, USA), a specific PI3K inhibitor [120], were proposed as a strategy to improve outcomes in BRAF-mutant patients with increased PI3K/AKT activity [111,120,121,122]. Preliminary encouraging activity of this regimen emerged from a phase Ib clinical trial [107], but further studies are needed to assess the role of alpelisib in mCRC patients.

### 3.3. NTRK Fusions

NTRK, also called tropomyosin receptor kinases TrkA, TrkB and TrkC, are transmembrane proteins encoded by three genes (NTRK1, NTRK2 and NTRK3) [123]. Stimulation of the receptors results in the activation of various signaling pathways responsible for differentiation, proliferation and survival of neuronal cells [123]. NTRK gene fusions lead to the development of chimeric rearrangements in TrkA, TrkB and TrkC, causing constitutive kinase activation [124]. NTRK fusions act as oncogenic drivers and may represent a potential target for therapeutic purposes [125], even in mCRC [126]. Although a wide comprehensive analysis reported less than 1% of NTRK gene fusions in unselected patients with mCRC, the occurrence increased to 5% and 15% in MSI-H and MSI-H/RAS/BRAF wt cancers, respectively [127]. In MLH1-deficient patients, an even higher rate (44%) of NTRK fusion rearrangements was described [128]. Overall, NTRK fusions in patients with mCRC more often correlated with right sidedness, female sex, elderly age, poor prognosis and resistance to anti-EGFR MoAbs [127,129].

#### 3.3.1. TRK Inhibitors

Two NTRK inhibitors were developed recently, namely larotrectinib (Vitrakvi^®^; Bayer Healthcare Pharmaceuticals, Berlin, Germany) (Figure 1), which was approved by the FDA in 2018 [130], and entrectinib (Rozlytrek^®^; Genentech Inc.) (Figure 1), which was approved by the FDA in 2019 [131]. In contrast to entrectinib, a pan-TRK also inhibiting ROS1 and ALK and able to bypass the blood-brain barrier [132,133], larotrectinib acts as a selective inhibitor of all three NTRK proteins and was more extensively evaluated in patients with cancer [134]. Both drugs exhibit efficacy in solid tumors characterized by NTRK gene fusions, including CRC [132,135,136,137,138].

#### 3.3.2. Overcoming Resistance to NTRK Inhibitors

Two mechanisms of resistance to NTRK inhibitors have been described. First, the so-called “on-target” mechanism induces amino acid substitutions in three regions of the NTRK kinase domain: the solvent front, the gatekeeper residue, or the xDFG trigger (Figure 5).

These mutations may account for resistance to NTRK inhibitors by sterically interfering with binding of the inhibitor or by altering the affinity of the ATP-binding site. Next-generation NTRK inhibitors, characterized by increased activity against wt TrkA, TrkB and TrkC were designed to overcome “on-target” resistance [129].

The second mechanism of resistance (“off-target” phenomenon) occurs due to genomic alterations involving other receptor tyrosine kinases or downstream pathways such as MET amplification, BRAF V600E or KRAS mutations (Figure 5). The combination of different TKIs could help to overcome this mechanism of resistance [129]. Indeed, the efficacy of dual NTRK and MET blockade in patients with NTRK fusion-positive cancer that developed MET amplification-driven resistance to a first-generation TRK inhibitor was evaluated in experimental models. By treating with MAPK targeted therapy, administered alone or in combination with TRK inhibition, disease control was re-established [139]. However, overcoming resistance to NTRK inhibitors remains a challenge.

### 3.4. Multi-Kinase Inhibitors

Beside the aforementioned TKIs, multi-kinase TKIs also entered in the therapeutic armamentarium, among these regorafenib (Stivarga^®^; Bayer Pharma AG, Berlin, Germany), a compound active against several protein kinases related to angiogenesis (e.g., VEGFR-1/2/3), oncogenic pathways such as KIT, BRAF and RET, the metastatic process and immunosuppressive activity of the tumor microenvironment [140]. In the randomized phase III CORRECT trial, regorafenib was compared to placebo in mCRC treated with all standard approved drugs, showing a modest OS improvement of 1.4 mo (HR: 0.77, *p* = 0.052) [141]. Only a few studies evaluated regorafenib in early lines in combination with chemotherapy, finding an acceptable toxicity profile [142] but no improvement in RR compared to historical controls [143]. As a consequence, the FDA and EMA approved regorafenib for mCRC previously treated with fluoropyrimidines, OX and IRI-based chemotherapy, anti-VEGF MoAbs and if KRAS wt anti-EGFR MoAbs [144]. Other anti-angiogenic TKIs (e.g., sorafenib, sunitinib, vandetanib and valatinib) are available, but data in mCRC are still preliminary and conflicting in terms of efficacy [140,145,146,147,148].

Unfortunately, to date, no specific predictive biomarkers have been identified to better select patients who may benefit from these agents.

## 4. Targeting Angiogenesis

Angiogenesis, or better neo-angiogenesis (a cancer-driven phenomenon other than what occurs in early phases of lifetime of the individuals), is a phenomenon in which new vessels develop either de novo or from existing vasculature, playing an important role in tumor initiation, growth and metastasis. This process is regulated by pro-angiogenic and anti-angiogenic factors, such as VEGF, fibroblast growth factor, transforming growth factor-α and β (TGF-α, TGF-β), platelet-derived endothelial cell growth factor and angiopoietins [149,150,151]. These factors, arising from cancer or stromal cells, stimulate tumor growth and metastasis [149,150,151]. The VEGFs, the first factors against which MoAbs have been directed, include five members: VEGF-A, B, C and D and placental growth factor. VEGF-C and D are not involved in vessel angiogenesis but in lymphoangiogenesis, so they have not been considered a putative target [152,153].

VEGF expression is increased by tissue hypoxia, growth factors and oncogenes (e.g., c-Src). The activated VEGFs bind VEGFR-2 and promote tumor angiogenesis by increasing permeability of post-capillary venules, leading to release of the plasma proteins (i.e., fibrinogen and clotting factors) into the extracellular matrix [154,155]. Afterwards, fibrinogen is converted to fibrine, which promotes endothelial cell migration and proliferation. In addition, VEGFR-2 plays a mitogenic action on endothelial cells, leading to endothelial proliferation by activating members of MAPK and protein kinase C pathways [154,156].

In first-line treatment, anti-angiogenic drugs in combination with chemotherapy represent an important treatment option in patients exhibiting RAS family mutations. The first anti-angiogenic drug was bevacizumab (Avastin^®^; Genentech/Roche) (Figure 1), an IgG1 humanized MoAb against the VEGF-A ligand. The initial FDA and EMA approval of bevacizumab in the treatment of mCRC resulted from pivotal phase I and II AVF2107 trials, showing the superiority of bevacizumab plus chemotherapy vs. chemotherapy alone, with an improvement of both PFS (10.6 mo vs. 6.2 mo, *p* < 0.001) and OS (20.3 mo vs. 15.6 mo, *p* < 0.001) [156]. However, subsequent phase III trials with bevacizumab plus FOLFOX, FOLFIRI or FOLFOXIRI did not homogeneously confirm an advantage in OS compared to chemotherapy, showing that benefit is irrespective of RAS genes mutational status [157,158,159,160,161]. Also, second-line treatment of bevacizumab plus FOLFOX was superior compared to FOLFOX, with longer PFS (7.3 mo vs. 4.7 mo, *p* < 0.001) and OS (12.9 mo vs. 10.8 mo, *p* = 0.0011) favoring the experimental arm [162]. Thus, the FDA and EMA recommend bevacizumab in combination with fluoropyrimidines plus OX or IRI for patients with RAS genes or BRAF mutations for which anti-EGFR MoAbs cannot be administered [163]. In addition, the EMA recommends bevacizumab in combination with any chemotherapy regimen that includes a fluoropyrimidine [164].

Two other anti-angiogenic drugs were developed for mCRC. Aflibercep is a soluble decoy receptor that binds VEGF-A, B and placental growth factor. In patients previously treated with OX-based chemotherapy +/− bevacizumab, the addition of aflibercept (Figure 1) to FOLFIRI conferred a modest OS advantage (13.50 mo vs. 12.06 mo, HR: 0.817, *p* < 0.0032) compared to chemotherapy alone [18]. In the RAISE trial, ramucirumab, a recombinant human IgG1 MoAb that binds to VEGFR-2 and acts as an antagonist to VEGF-A, VEGF-C and VEGF-D, was added to FOLFIRI in patients that progressed to FOLFOX plus bevacizumab (Figure 1). An OS improvement (13.3 mo vs. 11.7 mo, HR: 0.844, *p* = 0.0219) was reported in the experimental arm [117]. These trials with post-hoc analyses suggest that either continuation of anti-angiogenic agents beyond progression is an effective strategy and the choice of second-line anti-VEGF MoAb can be done taking into account first-line treatment PFS [116,117].

Nowadays, no routine biomarkers are used to select patients who may benefit from anti-angiogenic MoAbs. Circulating tumor cells, free nucleic acids, microRNAs (miRNAs), other angiogenesis markers and functional imaging (i.e., diffusion-weighted imaging, fluorodeoxyglucose positron emission tomography and dynamic contrast-enhanced magnetic resonance) have shown promising potential as predictive biomarkers of outcome [165]. Moreover, it has been proposed that early developing hypertension may be a predictive marker for the efficacy of bevacizumab [166,167,168].

### Overcoming Resistance to Anti-Angiogenic Agents

Some patients derive no or little benefit from VEGF inhibitors. The primary or secondary resistance to these agents is due to alternative angiogenic pathways. In cancer, the most relevant markers involved in anti-VEGF resistance are angiopoietin-2 (Ang2), fibroblast growth factor 2, IL-1, platelet-derived endothelial cell growth factor and TGF-β (Figure 6) [169].

Among these, only Ang2 has been clearly associated with resistance to anti-angiogenic agents in mCRC. Ang2 is involved in a crucial pathway for blood vessel development, remodeling and vascular permeability. It has been shown that high levels of Ang2 in the serum of patients with mCRC correlates with resistance to bevacizumab, suggesting that drugs targeting Ang2 could be able to restore the efficacy of anti-VEGF compounds [170].

In mCRC, vanucizumab (development code RG7221^®^; Genentech/Roche), a bi-specific MoAb binding both Ang2 and VEGF-A, has been evaluated in a randomized phase II trial in combination with FOLFOX vs. bevacizumab plus FOLFOX in chemo-naïve patients with mCRC. Unfortunately, the experimental arm failed to improve PFS, and increased anti-angiogenic toxicity was observed when compared to the standard arm. Based on these negative findings and having understood that Ang2 is not a relevant therapeutic target following pharmacokinetic and pharmacodynamic analysis, the clinical development of vanucizumab was discontinued [171,172].

Another mechanism of resistance to bevacizumab emerges in the anti-EGFR rechallenge strategy (Figure 6) (see the previous section “Overcoming resistance to anti-EGFR MoAbs”).

## 5. Immunotherapy

The efficacy of chemotherapy and targeted therapy in mCRC treatment has reached a plateau [173]. Despite progress, prognosis remains poor, and the OS does not exceed 30 mo [174].

Preclinical and clinical trials have demonstrated that immune checkpoint inhibitors (ICIs) improve treatment responses in many malignancies such as melanoma, renal cancer and lung cancer. ICIs, initially conceived to treat patients with advanced disease, have also been evaluated in neo- and adjuvant settings [175,176,177]. In contrast to traditional cytotoxic chemotherapies, immunotherapy acts by promoting the host immune system against cancer cells [173]. ICI drugs target the co-inhibitory receptors present on T cells, such as cytotoxic T lymphocyte-associated protein-4 (CTLA-4) and programmed cell death-1 (PD-1) or their ligand (PD-L1) expressed on cancer cells, by selectively blocking their interaction. This leads immune cells to attack cancer cells and establish a long-lasting immunity [178,179,180,181,182]. Objective RRs of ICIs range from 15% to 50%, depending on cancer type [183,184,185,186]. Highly mutated cancer cells stimulate several peptide antigens which are subsequently expressed on their major histocompatibility complex class I molecules. This induces the T cells to recognize and destroy them.

### 5.1. ICIs

CRCs are subdivided into groups. First, those characterized by deficiency in the mismatch repair system (dMMR or MSI-H), accounting for 15% of CRC and up to 5% of mCRC with respect to mismatch repair proficient (pMMR) or microsatellite-stable (MSS) or MSI-L, accounting for 85% of CRC and more than 95% of mCRC [187,188,189,190]. dMMR-MSI-H CRCs present a very high tumor mutation burden (TMB) as a consequence of the failure of the MMR genes in repairing replication-associated errors. The high number of DNA alterations leads to an increase of somatic mutations and then to the production of a number of tumor neoantigens, allowing a more favorable recognition of patients’ immune systems and consequently to a better response to ICIs [189,191,192,193].

In contrast to dMMR-MSI-H, pMMR-MSI-L/MSS CRCs exhibit a very low level of tumor mutations, which is believed to be the main reason for unresponsiveness to immunotherapy [190]. Recent studies have suggested that pMMR-MSI-L/MSS CRCs presenting a high TMB are associated with better clinical responses to immunotherapy [194,195,196,197,198], thus opening the opportunity to explore TMB as a potential predictive marker. The FDA approved the anti PD-1 pembrolizumab (Keytruda^®^; Merck Sharp, Kenilworth, NJ, USA) (Figure 1) [199] and nivolumab [200] (Opdivo^®^; Bristol Myers Squibb, New York, NY, USA) (Figure 1), with or without the anti-CTLA-4 ipilimumab (Yervoy^®^; Bristol Myers Squibb) (Figure 1), in the treatment of dMMR-MSI-H patient with mCRC. In addition, the combination of PD-1 and CTLA-4 inhibitors (e.g., nivolumab and ipilimumab [201] or new molecules under approval such as botensilimab and balstilimab [202]) was found to be effective in the treatment of MSS mCRC that never had liver metastases. In the REGONIVO trial, regorafenib in combination with nivolumab produced an encouraging overall RR of 33% in MSS patients with CRC [203,204].

#### 5.1.1. ICI Resistance

In contrast with dMMR-MSI-H patients, pMMR-MSI-L CRCs present several mechanisms of intrinsic resistance to ICIs. In addition to low TMB and subsequent lack of tumor antigens, the activation of β-catenin signaling, which is frequently observed in MSS CRCs, confers resistance to immune checkpoint blockade by excluding T cells within the tumor microenvironment [205,206]. The resistance to ICIs observed in MSS cases was reported also when ICIs (i.e., atezolizumab) have been combined with another targeted inhibitor such as cobimetinib (a MEK inhibitor) [207]. Moreover, TGF-β has been associated with immune evasion as well. This mechanism, as reported by previous studies, is partially involved in the resistance to immunotherapy in MSS CRC by enhancing cancer-associated fibroblasts to promote intratumoral fibrosis [208,209]. Immunotolerance induced by liver metastasis may also be a cause of immunotherapy resistance in pMMR-MSI-L/MSS mCRCs, which are indeed more associated with liver metastasis as compared to dMMR/MSI-H [210,211].

**Table 1 jcm-11-07523-t001:** Targeted therapies and the relevant mechanisms for overcoming resistance in metastatic colorectal cancer.

Therapy	Molecular Target	Resistance Mechanisms	How to Overcome Resistance
Anti-EGFR MoAbs (cetuximab, panitumumab)	EGFR	(1) Mutations in EGFR molecular pathway: RAS family, BRAF; (2) Mutations or amplification of HER2 gene; (3) BRAF V600E mutation; (4) PIK3CA exon 20 mutations and PTEN loss; (5) MET amplification; (6) Levels of AREG and EREG gene expression.	(1) Simultaneous or sequential inhibition of the pathways involved in the altered markers leading to resistance (e.g., administration of BRAF V600E inhibitors) [60]; (2) Inhibition of EGFR downstream effectors such as mitogen-activated protein kinases (MEK or MAP2K) or mechanistic target of rapamycin (mTOR) [61,62,63,64]; (3) Anti-angiogenic (reversion of tumors from KRAS mut to KRAS wt) [67]; (4) Second-generation anti-EGFR MoAbs [41,68,69].
Anti-HER2 MoAbs (trastuzumab, pertuzumab)	HER2	(1) Mutations in RAS family, PIK3CA and BRAF.	(1) Change of the original anti-HER2 doublet to a drug combination including a multi-kinase inhibitor (e.g., from pertuzumab plus trastuzumab to trastuzumab plus lapatinib) [22].
BRAF TKIs (vemurafenib, encorafenib)	BRAF	(1) Adaptive feedback signaling networks (RAS activity will increase by generating BRAF inhibitor-resistant RAF dimers. The increase in RAS activity is thought to be driven primarily by EGFR.	(1) Combination of BRAF and EGFR inhibition (e.g., vemurafenib plus irinotecan and cetuximab; encorafenib plus cetuximab) [103,107,108,116,117,138]; (2) Combination of BRAF and MEK inhibition (encorafenib plus binimetinib) [104,109,110].
NTRK inhibitors (larotrectinib, entrectinib)	NTRK	(1) “On-target” mechanism inducing amino acids substitutions in three regions of the NTRK kinase domain; (2) Genomic alterations involving other receptor tyrosine kinases or downstream pathways: MET amplification, BRAF V600E mut, KRAS mut.	(1) Next-generation NTRK inhibitors able to overcome the mutations described for the “on target” mechanism [129]; (2) Inhibition of two molecular target (e.g., MAPK and NTRK) [139].
Multi-kinase inhibitors (regorafenib, sorafenib, sunitinib, vandetanib, valatinib)	Different proteins involved in angiogenesis and oncogenic pathway simultaneously	No data available	No data available
Anti-angiogenic drugs (bevacizumab, aflibercept, ramucirumab)	VEGFs, VEGFRs	(1) High levels of Ang2 in serum of mCRC patients; (2) Appearance of EGFR mutations.	(1) Administration of vanucizumab a bispecific MoAb binding both Ang2 and VEGF-A [171,172]; (2) Anti-EGFR rechallenge [65,66,67].
Immunotherapy (pembrolizumab, nivolumab, ipilimumab)	PD-1, PD-L1, CTL4	(1) MMR-MSI-L CRCs; (2) Low TMB; (3) Constitutive PD-L1 expression; (4) Lack of the genes for response to IFN-γ.	(1) Concomitant administration of drugs targeting MEK, STING, WNT or oncolytic viruses [208,209,210,211,212,213,214,215,216,217,218]; (2) Combination of different ICIs (preliminary data) [22,26].

Abbreviations: ICI: Immune checkpoint inhibitors; MoAb: Monoclonal antibodies; mut: Mutated; TKIs: Tyrosine kinase inhibitors; *wt*: Wildtype; mCRC: Metastatic colorectal cancer; MMR: Mismatch repair; MSI-L: Microsatellite instability low; TMB: Tumor mutation burden; IFN-γ: Interferon gamma; PD-L1: Programmed cell death ligand-1; PD-1: Programmed cell death-1; CTL4: Cytotoxic T lymphocyte associated protein-4; EGFR: Epidermal growth factor receptor; HER2: Human epidermal growth factor 2; VEGFR: Vascular endothelial growth factor receptor; VEGF: Vascular endothelial growth factor; MEK: Mitogen-activated protein kinase; MAP2K: Mitogen-activated protein kinase; KRAS: Kirsten rat sarcoma virus RAS; NTRK: Neurotrophic tyrosine receptor kinases; Ang2: Angiopoietin-2; STING: Stimulator of interferon genes; CRC: Colorectal cancer.

#### 5.1.2. Overcoming ICI Resistance

Novel combination strategies may be considered to overcome intrinsic resistance to immunotherapy in pMMR-MSI-L/MSS mCRCs. The concomitant administration of drugs targeting the mitogen-activated protein/extracellular signal-regulated kinase (MEK) [212], stimulator of interferon genes [213,214,215] or WNT [216] or oncolytic viruses [217,218,219,220,221] are putative ways to overcome resistance. A recent study [222] demonstrated that targeting IL-17A, which stimulates tumorigenesis, angiogenesis and metastasis in CRC [223], may be promising to sensitize MSS CRC to ICIs. In fact, previous reports suggested that a T helper 17 and IL-17 signature correlated with poor prognosis in MSS patients with mCRC [224,225], and a recent study revealed that the activation of IL-17A signaling correlated to the failure of anti-PD-1 therapy in patients with MSS mCRC [226]. However, preclinical and clinical studies are needed to validate the hypothesis generated by Liu et al. [222]. Overall, immunotherapy resistance can arise as a consequence of tumor cell-intrinsic and microenvironmental factors. As for PD-1/PD-L1 blockade resistance, the constitutive PD-L1 expression in tumor cells as well as the lack of tumor antigens, defective antigen presentation, activation of oncogenic pathways, mutations in interferon gamma signaling and changes in the tumor microenvironment are the most important factors associated with immunotherapy resistance [227].

The CTLA-4 immune checkpoint, the first inhibitory immune checkpoint identified, is dependent on the co-stimulatory molecule B7 to induce its response. CTLA-4 blockade leads to enhanced T cell function, and T cells interact with CTLA-4 blockade through interferon gamma. The main mechanism of resistance to CTLA-4 blockade involves the lack of the genes for response to interferon gamma by tumor cells [228].

A possible solution to overcome resistance to immunotherapy is the combination of ICIs. Previous clinical trials reported that anti-CTLA-4 plus anti-PD-1 antibodies lead to an improved RR compared to a single ICI compound. However, combination of ICIs correlated with a worsening toxicity profile. Indeed, the reported adverse events rate with single-agent CTLA-4 is 27.3%, with PD-1 alone being 16.3% and 55% for the combination [229]. Recently, a small study conducted in treatment-refractory melanoma showed efficacy of fecal microbiota transplantation in reprogramming the tumor microenvironment, leading to increased cancer immunogenicity [229]. However, there is no proof that this strategy could be worthwhile in mCRC.

## 6. Extracellular Vesicles and miRNA

Extracellular vesicles (Evs) are a heterogeneous population of nano-sized membrane-delineated vesicles [230], classified as exosomes (30–100 nm), microvesicle (100 nm–1µm) and apoptotic bodies (50 nm–5 µm) [231]. Evs have been demonstrated to be an important mechanism of intercellular communication [232]. The cellular interchange occurs through the release of large amounts of nucleic acids, mRNA, miRNA, cytokines/chemokines, angiogenic factors, extracellular matrix remodeling factors as well as tumor microenvironment modulating factors during carcinogenesis [233,234,235]. Moreover, additional studies have also shown that Evs can alter the function and phenotype of recipient cells [230,236]. Concerning CRC, these alterations can either affect the regulation of several cellular processes, modulate the tumor microenvironment or enhance tumor cell proliferation and transformation [230,236,237].

Evs have been suggested to interfere with regulation of the immune system process by activating the antigen-specific CD4+ or CD8+ T cell through antigen presentation [238,239]. The intrinsic feature of Evs to regulate T cells allows cancer-associated Evs to promote tumor progression by facilitating immune modulation and suppression within the tumor microenvironment through remodeling of tumor-stromal interactions, transfer of genetic materials (such as proteins, mRNA and non-coding RNAs), release of myeloid-derived suppressor cells and immune suppressive molecules [240]. Evs have also been associated with DR in patients with CRC. Cytotoxic drugs seem to be sequestered in intracellular vesicles by exosomes and subsequently ejected in order to negate drug effects within the cells [241,242,243]. Although Evs DR was mainly associated to standard chemotherapy [244], a study conducted on CRC lines demonstrated an association with cetuximab resistance by downregulation of PTEN and subsequent increased phosphorylation of AKT levels [245].

In addition to Evs, miRNAs were associated to chemotherapy resistance by promoting stemness in cancer stem cells, leading to atypical morphology. The study conducted by Ren et al. [246] demonstrated that patients having high levels of miR-196b-5p had poor survival due to the targeting of immune regulators of the STAT3 signaling pathway. As a consequence, STAT3 is activated and promotes stemness and chemoresistance. However, to the best of our knowledge and excluding chemotherapy, no further target DR was found concerning miRNA.

## 7. Conclusions

With the advent of molecular-targeted drugs and immunological agents in addition to chemotherapy, the prognosis of patients with mCRC has improved. The treatment algorithm of mCRC patients is tailored according to three major points, i.e patient characteristics (performance status, comorbidities, age, previous adjuvant treatment, preferences, acceptance of toxicities and expectations); tumour characteristics (metastatic burden, potential resectabity and sites of metastasis, primary tumour location) and molecular profile (RAS genes/BRAF status, microsatellite instability and, eventually, HER2 overexpression and NTRK rearrangement). First-line therapy is a key determinant of outcome using doublets o triplets in combination with anti-EGFR MoAbs or bevacizumab. Two third of patients will receive second line therapy, and the regimen choice depends on the systemic therapies given in first line. For patients who can tolerate further treatment lines, maintenance of quality of life and performance status are crucial other than disease control. Nevertheless, the benefit obtained from these drugs is temporally limited, mostly due to development of resistance mechanisms. Better understanding of the mechanisms underlying resistance and possible solutions to overcome them may confer further improvement in patient prognosis.

## Figures and Tables

**Figure 1 jcm-11-07523-f001:**
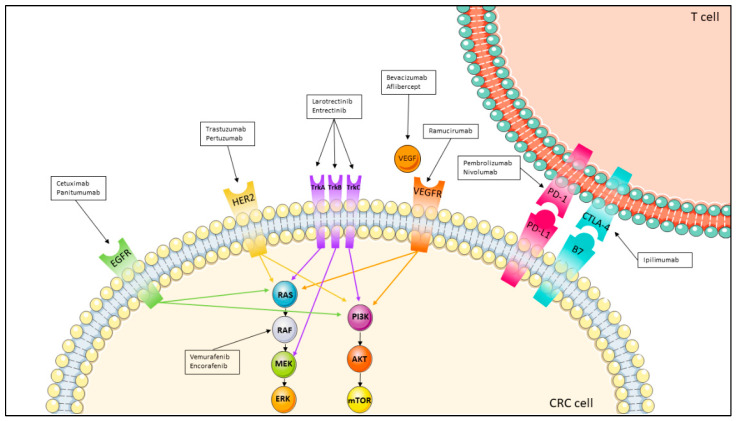
Targeted agents in metastatic colorectal cancer. Molecular pathways and the respective targeted agents involved in metastatic colorectal cancer treatment. Abbreviations: AKT: Protein kinase B; CRC: Colorectal cancer; CTL-4: Cytotoxic T-lymphocyte antigen 4; EGFR: Epidermal growth factor receptor; ERK: Extracellular-signal regulated kinase; HER2: Human epidermal growth factor 2; MEK: Mitogen-activated protein kinase; mTOR: Mechanistic target of rapamycin; PD-1: Programmed cell death protein 1; PD-L1: Programmed cell death protein ligand-1; PI3K: Phosphatidylinositol-4,5-bisphosphate 3-kinase; RAS: RAS proteins; Tcell: lymphocyte T; TrkA: Tyrosine kinase receptor A; TrkB: tyrosine kinase receptor B; TrkC: tyrosine kinase receptor C; VEGF: Vascular endothelial growth factor; VEGFR: Vascular endothelial growth factor receptor.

**Figure 2 jcm-11-07523-f002:**
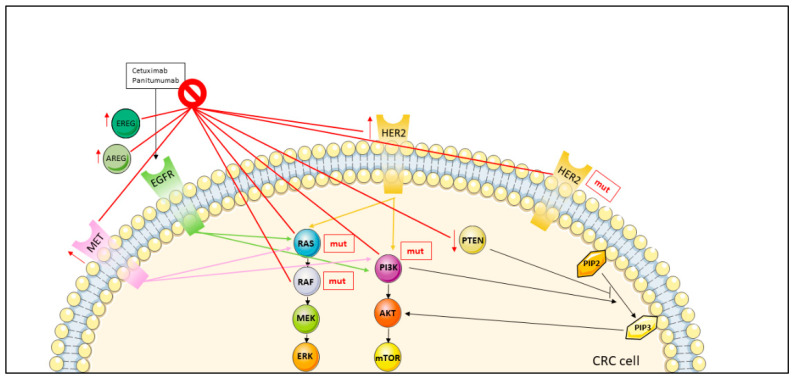
Mechanisms of resistance to anti-EGFR antibodies in metastatic colorectal cancer. The prohibition symbol stays for resistence to the therapy, the upward red arrow for higher number of copies (overexpression), and the red downward for underexpression. Mechanisms of resistance are signed with red lines. Abbreviations: AKT: Protein kinase B; AREG: Amphiregulin; CRC: colorectal cancer; EGFR: Epidermal growth factor receptor; EREG: Epiregulin; ERK: Extracellular-signal regulated kinase; HER2: Human epidermal growth factor 2; MEK: Mitogen-activated protein kinase; MET: Mesenchymal epithelial transition protein; mTOR: Mechanistic target of rapamycin; mut: mutated; PIK3CA: Phosphatidylinositol-4,5-bisphosphate 3-kinase; PIP2: Phosphatidylinositol 4,5-bisphosphate; PIP3: Phosphatidylinositol 3,4,5-triphosphate; PTEN: Phosphatase and tensin homolog; RAS: RAS proteins.

**Figure 3 jcm-11-07523-f003:**
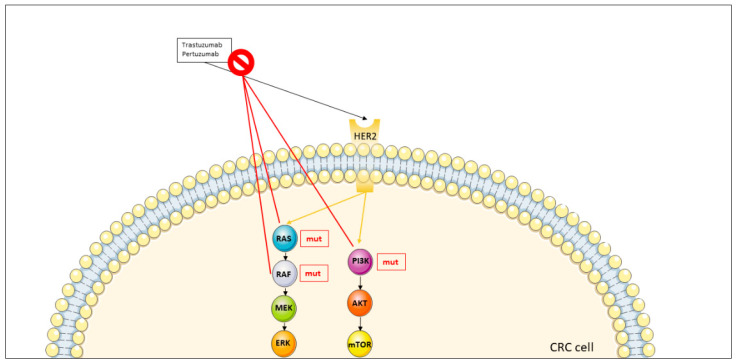
Mechanisms of resistance to anti-HER2 antibodies in metastatic colorectal cancer. The prohibition symbol stays for resistence to the therapy and the mechanisms of resistance are signed with red lines. Abbreviations: AKT: Protein kinase B; CRC: colorectal cancer; ERK: Extracellular-signal regulated kinase; HER2: Human epidermal growth factor 2; MEK: Mitogen-activated protein kinase; mTOR: Mechanistic target of rapamycin; mut: mutated; PI3K: Phosphatidylinositol-4,5-bisphosphate 3-kinase; RAS: RAS proteins.

**Figure 4 jcm-11-07523-f004:**
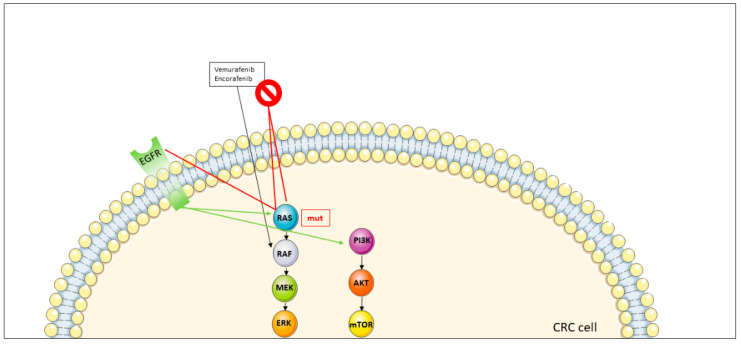
Mechanisms of resistance to BRAF inhibitors in metastatic colorectal cancer. The prohibition symbol stays for resistence to the therapy and the mechanisms of resistance are signed with red lines. Abbreviations: AKT: Protein kinase B; CRC: colorectal cancer; EGFR: Epidermal growth factor receptor; ERK: Extracellular-signal regulated kinase; MEK: Mitogen-activated protein kinase; mTOR: Mechanistic target of rapamycin; mut: mutated; PI3K: Phosphatidylinositol-4,5-bisphosphate 3-kinase; RAS: RAS proteins.

**Figure 5 jcm-11-07523-f005:**
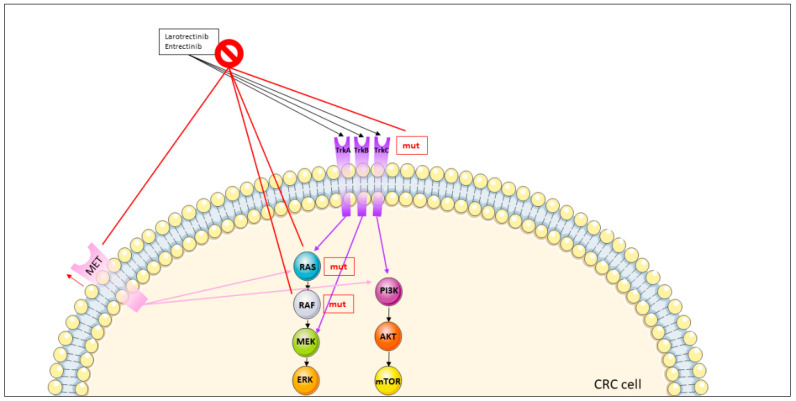
Mechanisms of resistance to NTRK inhibitors in metastatic colorectal cancer. The prohibition symbol stays for resistence to the therapy and the upward red arrow for higher number of copies (overexpression). The mechanisms of resistance are signed with red lines. The single abbreviation “mut” beside the three NTRK receptors (TrkA, TrkB and Trkc) indicates that the mutations can be either in TrkA or TrkB or Trkc. Abbreviations: AKT: Protein kinase B; CRC: colorectal cancer; ERK: Extracellular-signal regulated kinase; MEK: Mitogen-activated protein kinase; MET: Mesenchymal epithelial transition protein; mTOR: Mechanistic target of rapamycin; mut: mutated; PI3K: Phosphatidylinositol-4,5-bisphosphate 3-kinase; RAS: RAS proteins, TrkA: tyrosine kinase receptor A; TrkB: tyrosine kinase receptor B; TrkC: tyrosine kinase receptor C.

**Figure 6 jcm-11-07523-f006:**
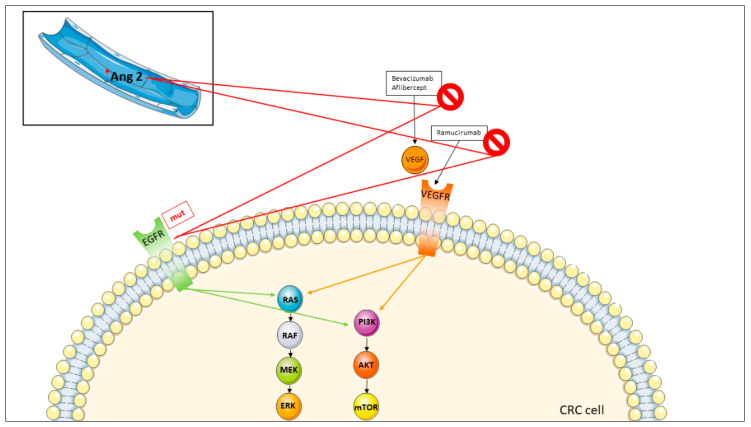
Mechanisms of resistance to anti-angiogenic drugs in metastatic colorectal cancer. The prohibition symbol stays for resistence to the therapy and the upward red arrow for higher number of copies (overexpression). The mechanisms of resistance are signed with red lines. Abbreviations: AKT: Protein kinase B; Ang: Angiogenin; CRC: Colorectal cancer; EGFR: Epidermal growth factor receptor; ERK: Extracellular-signal regulated kinase; MEK: Mitogen-activated protein kinase; mTOR: Mechanistic target of rapamycin; mut: mutated; PI3K: Phosphatidylinositol-4,5-bisphosphate 3-kinase; RAS: RAS proteins; VEGF: Vascular endothelial growth factor; VEGFR: Vascular endothelial growth factor receptor.

## Data Availability

Not applicable.

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
