# Peer review of "Recent and Future Strategies to Overcome Resistance to Targeted Therapies and Immunotherapies in Metastatic Colorectal Cancer"

_jcm, 2022, doi:10.3390/jcm11247523_

Round 1

Reviewer 1 Report

The submitted manuscript by Dr. Dazio and colleagues highlights advances in treatment strategy for metastatic colorectal cancer, potential mechanisms for acquired resistance and strategies to overcome resistance and improve responses. Overall, this review is very thorough, but a bit long. There was a review published earlier this year in another MDPI journal (cancers) with a similar topic, but not limited to metastatic colorectal cancer. I recommend the manuscript be revised for clarity and length, with specific attention to the items below.

11.      Include diagrams of resistance mechanisms (as did the authors in the ‘all CRC’ treatment review mentioned above: doi.org/10.3390/cancers14122928) to provide a visual representation of pathway overlap and current drugs to target them (alone or in combination).

22.      Shorten the article. There are many shared pathways of resistance that can be described using figures (see above), rather than repeated for each molecular target section.

33.      Edit for grammar/linguistic issues. There are several places in the manuscript where the word(s) used does not have the connotation that I believe the authors intended. I have highlighted a couple examples below (though not an exhaustive list):

a.      In the abstract, the second-to-last sentence currently reads: ‘In this review we describe the most relevant targeted therapies and immunotherapies and expand on the reasons for resistance to the different approved or underdeveloped targeted drugs.’ I am wondering whether the authors had ‘molecularly targeted drugs currently under development’ in mind? Otherwise, I am not sure what this sentence means, as ‘underdeveloped’ is more often used to describe ideas (hypotheses), or biological processes (e.g., puberty).

b.      On page 12, the heading “4.1. Overcoming resistance to anti-angiogenetic agents” – I believe this is a typo and meant to be anti-angiogenesis or anti-angiogenic agents.

                                                    i.     Two sentences later, the text currently reads: ‘The primary or secondary resistance to these agents is owned by alternative angiogenic pathways.’ Perhaps a more accurate way to say this is ‘due to’?

4.  Optional: It may be useful to include a discussion on differences between treatment naïve metastatic cancer and metastatic cancers treated with multiple lines of therapy that become resistant. Such a discussion may add more interest and distinguish this review from others.

Author Response

Dear reviewer 1,

please find attached our point by point response letter.

Kind regards

Samantha Epistolio

Author Response

Dear reviewer 2,

please find attached our point by point response letter.

Kind regards

Samantha Epistolio

Reviewer 3 Report

The review is an excellent manuscript summarizing the actual literature on tumor strategies on the development of resistance in colorectal cancer. 

In order to improve the manuscript further I would suggest to add 2 to 3 figures presenting the described mechanisms graphically.  

Author Response

Dear reviewer 3,

please find attached our point by point response letter.

Kind regards

Samantha Epistolio

Round 2

Reviewer 1 Report

Thank you for your attention and responses to my previous comments and especially for constructing and adding figures to the manuscript. There are a couple adjustments I would like to suggest.

1.      Since many of the resistance pathway alterations are shared, it may be helpful to combine several of the resistance figures (like the pathway figure 1) and distinguish the treatment by font color or other labeling. This would be more concise and limit the figure count.

2.      Inconsistent definition of acronyms. There are a couple gene/protein acronyms that are defined differently in different areas of the manuscript. Specifically, mTOR is referred to as ‘mammalian Target of Rapamycin’ in table 1 and line 290 on page 8, but ‘Mechanistic target of rapamycin’ in the figure legends. Similarly, RAS is sometimes defined and sometimes not (see line 103 on page 4 and line 203 on page 6).

Author Response

RESPONSE TO REVIEWER 1

We would like to thank the reviewer for the additional comments. Please find below, in red, the detailed point-by-point reply.

  1. Since many of the resistance pathway alterations are shared, it may be helpful to combine several of the resistance figures (like the pathway figure 1) and distinguish the treatment by font color or other labeling. This would be more concise and limit the figure count.

Thank you for your kind opinion. We tried to combine all the resistance mechanisms` figures in a single one, however we obtained a cahotic and incomprehensible figure that is really difficult to be read because the different resistances are not distinguishable. In addition we believe that for the reader it may be easier to have the pictures close to the respective text due to the length of the review. However, we will obey to editor’s choice: if the editor prefer to have a single picture including all the mechanism of resistance, we will prepare the figure even if it will result cahotic.

  1. Inconsistent definition of acronyms. There are a couple gene/protein acronyms that are defined differently in different areas of the manuscript. Specifically, mTOR is referred to as ‘mammalian Target of Rapamycin’ in table 1 and line 290 on page 8, but ‘Mechanistic target of rapamycin’ in the figure legends. Similarly, RAS is sometimes defined and sometimes not (see line 103 on page 4 and line 203 on page 6).

Thank you for this precisation: we have changed the text according to your suggestions.

Reviewer 2 Report

No further comments

Author Response

Thank you for having accepted the revisions.